# Evolution of Reproductive Traits and Implications for Adaptation and Diversification in the Yam Genus *Dioscorea* L.

**Min Chen** [1], **Xiaoqin Sun** [1], **Jia-Yu Xue** [2], **Yifeng Zhou** [1] **and Yueyu Hang** [1,*]

[1] Institute of Botany, Jiangsu Province and Chinese Academy of Sciences, Nanjing 210014, China; chenmin@cnbg.net (M.C.); xiaoqinsun@cnbg.net (X.S.); njgzhou@163.com (Y.Z.)
[2] College of Horticulture, Academy for Advanced Interdisciplinary Studies, Nanjing Agricultural University, Nanjing 210014, China; xuejy@njau.edu.cn
[*] Correspondence: hangyueyu@cnbg.net; Tel.: +86-025-84347052

**Abstract:** *Dioscorea* is a pantropical monocotyledonous genus encompassing several well-known tuber crops and medicinal plants. It possesses remarkable morphological diversity, especially in reproductive characteristics, which are suggested to play important roles in species adaptation and diversification. Yet there have been few studies that consider the evolutionary pattern followed by these characters in this genus. In this study, the phylogenetic relationships among Chinese yams were reconstructed from five chloroplast and two mitochondrial DNA sequences. The evolutionary histories of bulbil possession, inflorescence architecture, the color of the male flowers and the degree of male flower opening were reconstructed. The results suggested that yam bulbils evolved after the divergence between *D.* sect. *Testudinaria* and other species of *Dioscorea* except for in *D.* sect. *Stenophora* and *D.* sect. *Apodostemon*. The evolutionary trend in the degree of male flower opening ranged from fully open to nearly closed. Male flowers with dark colors and panicles were shown to be derived in *Dioscorea*. These characteristics were found to be closely associated with the reproductive patterns and pollinating mechanisms of the *Dioscorea* species. The findings also shed light on the systematic relationships within this genus.

**Keywords:** phylogeny; character evolution; bulbil; inflorescence architecture; floral color; perianth opening degree



## 1. Introduction

*Dioscorea* L. is the largest genus of the Dioscoreaceae, comprising about 630 species [1], mainly distributed in tropical and subtropical areas, sometimes expanding to temperate regions [2]. Species of this genus are mostly dioecious vines with underground storage organs [3]. The edible starchy tuber makes it the third most important tropical tuberous crop globally [4], and the metabolite-rich rhizomes of some species are used as a source of pharmaceutical compounds [5]. Apart from its economic importance, *Dioscorea* is one of the most critical taxa in monocot systematics since it is laid near the basal position in the phylogenetic tree of monocotyledonous plants and has a number of similarities with dicots [6,7]. However, there are challenges in the taxonomic and systematic investigation of *Dioscorea* due to its great morphological diversity, dioecy and small flowers [2,8]. There are about 1600 taxonomic names attributed to *Dioscorea*, but most of them are considered synonyms [1], indicating the controversy on the circumscription of species boundaries. Moreover, the classification system of *Dioscorea* differs greatly among authors, and many of the proposed infrageneric taxa do not quite represent natural lineages [2]. Based on morphological and anatomic characteristics, *Dioscorea* species have been assigned to between 24 and 58 sections in traditional taxonomic studies [3,9–11]. Recent phylogenetic analyses based on molecular data sets, including cpDNA regions and nuclear genes, have led to the recognition of 10–11 main clades in this genus [2,12–15]. The results greatly simplified previous classification

systems and gave rise to a fundamental phylogenetic framework for understanding the infrageneric relationships and evolutionary history of *Dioscorea*. Despite extensive study of phylogeny and biogeography, research focusing on character evolution and its relation to species diversification and ecological success are extremely limited.

During angiosperm evolution, advances in vegetative and reproductive organs generated remarkable morphological diversity and were pivotal to niche adaptation and speciation [16]. Recent phylogenetic research has indicated some lineages of *Dioscorea*, such as the Madagascar group and *D.* sect. *Enantiophyllum*, have very short phylogenetic branches, resulting from radiation events [2,12,13,17]. It has been proposed that fast evolution over a short period may reflect responses to tremendous climate change or the rise of innovative characters [18]. *Dioscorea* species exhibit extreme morphological variations; for example, underground storage organs include branching rhizomes, perennial or annual tubers, cylindrical or globose tubers, right or left stem twining and seed wings extending from the apex, the base or around the whole seed. Previous studies revealed the distribution patterns of some of these traits, including tuber morphology, stem twining direction, seed wing shape [2], stem anatomy [8], leaf venation [19], pollen characters and chromosome numbers [15]. However, these studies mainly focused on identifying synapomorphies for infrageneric lineages rather than elucidating the evolutionary history of traits and their links with the wide range of environmental conditions in which the plants lived.

Burkill [11] suggested that adaptation to limiting rainfall was the most important factor underpinning species diversification in *Dioscorea*. The restricted distribution and fewer species in *D.* sect. *Stenophora,* compared with great diversity and pantropical distribution of other clades of *Dioscorea*, led Wilkin et al. [2] to propose that tubers play an important role in the origins of diversity in this genus. Investigations focusing on African *Dioscorea* species showed that the shift from forest to open grassland was associated with changes in tuber size and orientation that protect the plants from fires. Their stem habit shifted from twining to erect due to the lack of supporting vegetation, and seed wing morphology adapted to release at low height, requiring higher wind speeds for efficient dispersal [20]. In addition, changes in flower and fruit morphology were suggested to play key roles in the exposure to radiation of the Madagascan clade [2]. These morphological traits can greatly influence the pollination process and seed dispersal, further altering the fitness of different species in given conditions. However, the evolutionary pattern of reproductive characters in *Dioscorea*, especially floral traits, remain poorly understood.

The Himalayan-Hengduan Mountains are purported to be the center of origin and diversification for the yam genus, with a large proportion of endemic species [21]. There are 52 species, 1 subspecies and 9 varieties of *Dioscorea* in China, which show enormous variability in traits related to sexual and vegetative reproduction [22]. For instance, some species produce bulbils at the leaf axils, while others do not; the inflorescence may be spikes, racemes or panicles; the color of male flowers varies between white, yellow, green to orange, purple and so on; and the perianths of male flowers are completely open, half-closed or fully closed during blooming in different species (shown in Figure 1). These features are widely used to divide plants into infrageneric groups and define species. This group of *Dioscorea* species is, therefore, an excellent candidate for exploring character evolution and implications for species diversification and ecological adaptation. In this study, we reconstructed the phylogenetic relationships between 48 *Dioscorea* species using 5 chloroplast and 2 mitochondrial DNA markers. Based on the phylogenetic framework produced, we explored the evolutionary patterns of four reproductive characters to clarify the driving force for the diversification and adaptive evolution of this important angiosperm lineage.

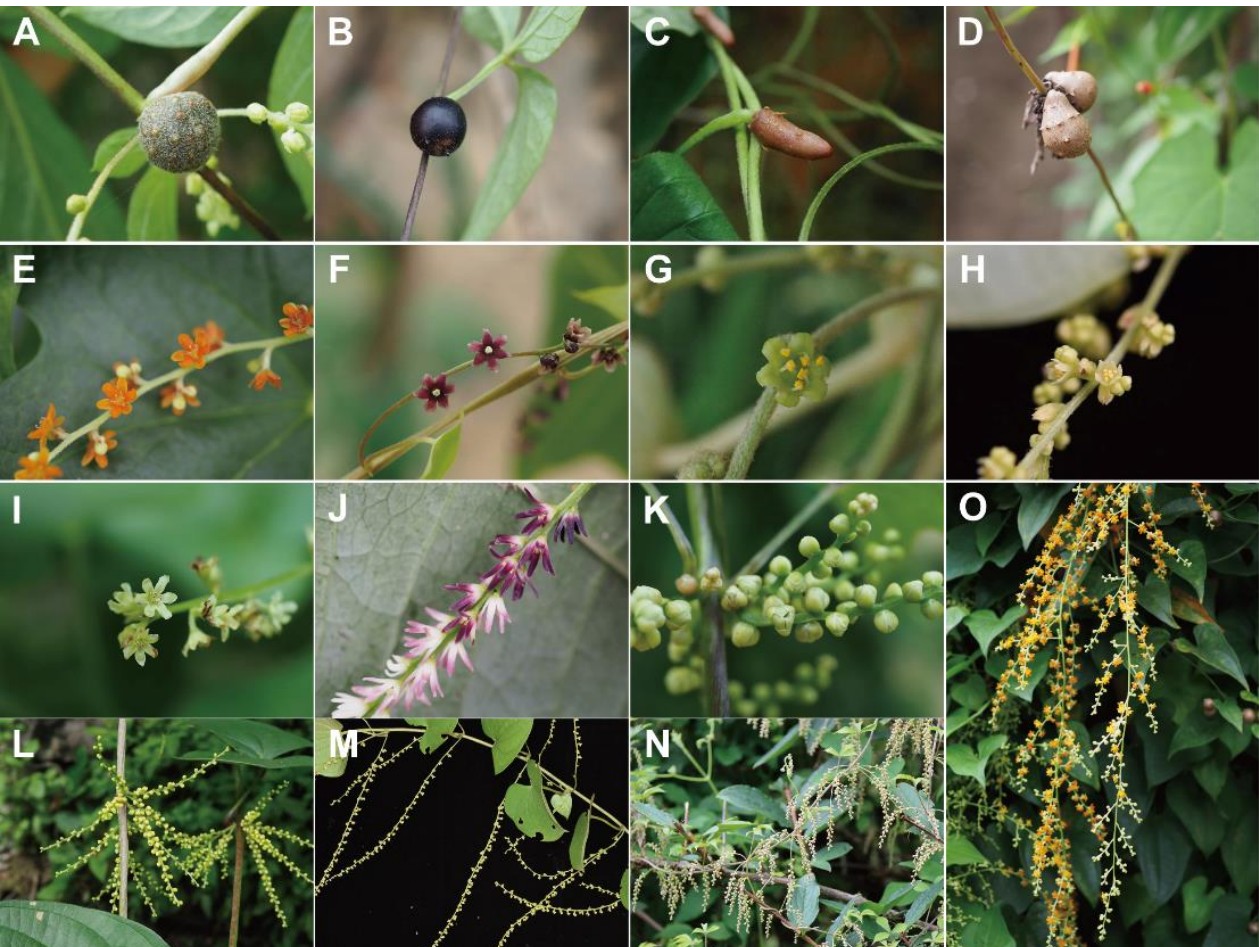

**Figure 1.** Characters related to reproduction showing morphological diversity of *Dioscorea*. (**A–D**) Bulbils. (**E–H**) Floral color. (**I–K**) Perianth opening degree. (**L–O**) Inflorescence architecture. (**A**) *D. kamoonensis*. (**B**) *D. melanophyma*. (**C,N**) *D. delavayi*. (**D,K**) *D. polystachya*. (**E,O**) *D. futschauensis*. (**F**) *D. zingiberensis*. (**G**) *D. yunnanensis*. (**H**) *D. subcalva*. (**I**) *D. tokoro*. (**J**) *D. bulbifera*. (**L**) *D.exalata*. (**M**) *D. panthaica.*

## 2. Materials and Methods

### 2.1. Sampling

A total of 48 *Dioscorea* taxa were sampled in this study, covering all sections distributed in China, except the monotypic section *D.* sect. *Stenocorea*. The samples consisted of 43 species, 1 subspecies and 4 varieties. Among them, 47 taxa were collected in the field and the vouchers were deposited in the Herbarium of the Institute of Botany, Jiangsu Province and the Chinese Academy of Sciences, China (NAS). The sequences of the remaining species, *D. wallichii* J. D. Hooker, were downloaded from GenBank. We selected *Tacca chantrieri* Andre (Dioscoreaceae), a representative of the genus closely related to *Dioscorea* according to the recently built phylogeny of the Dioscoreaceae [14], as the outgroup in our analyses. The geographical origin, voucher specimen information and GenBank accession numbers of all samples are listed in Table 1.

**Table 1.** Taxa used in this study with locality, voucher information and GenBank accession numbers.

| Species | Locality | Voucher | GenBank Accession No. | | | | | | |
|---|---|---|---|---|---|---|---|---|---|
| | | | *mat*K | *trn*L-F | *rbc*L | *psb*A-*trn*H | *rpl*36-*rps*8 | *nad*1 | *rps*3 |
| *Dioscorea nipponica* Makino | Lin'an, Zhejiang, China | NAS 0648570 | AY957600 * | DQ841308 | AF307455 * | GQ265171 | GQ265222 | GQ265123 | GQ265268 |
| *D. nipponica* subsp. *rosthornii* (Prain & Burkill) C. T. Ting | Tianshui, Gansu, China | NAS 0648571 | DQ974184 | DQ841309 | DQ408178 | GQ265172 | GQ265223 | GQ265122 | GQ265269 |
| *D. althaeoides* R. Knuth | Dêqên, Yunnan, China | NAS 0648572 | EU407548 | EU301741 | EU407550 | GQ265182 | GQ265233 | GQ265135 | GQ265281 |
| *D. tokoro* Makino | Anhua, Hunan, China | NAS 0648573 | DQ974186 | DQ841312 | DQ408180 | GQ265174 | GQ265225 | GQ265125 | GQ265271 |
| *D. zingiberensis* C. H. Wright | Mt. Hengshan, Hunan, China | NAS 0646476 | AY973831 * | DQ841318 | AY939889 * | GQ265154 | GQ265206 | GQ265105 | GQ265251 |
| *D. sinoparviflora* C. T. Ting, M. G. Gilbert & N. J. Turland | Lijiang, Yunnan, China | NAS 0648574 | DQ974179 | DQ841326 | DQ408171 | GQ265163 | GQ265212 | GQ265112 | GQ265258 |
| *D. deltoidea* Wall. ex Griseb. | Kunming, Yunnan, China | NAS 0648575 | EF614207 | DQ841305 | EF614218 | GQ265169 | GQ265220 | GQ265120 | GQ265266 |
| *D. panthaica* Prain & Burkill | Lijiang, Yunnan, China | Y. F. Zhou & B. C. Wu 200308015 | GQ265088 | GQ265291 | GQ265187 | GQ265184 | GQ265235 | GQ265136 | GQ265283 |
| *D. biformifolia* C. Pei & C. T. Ting | Mt. Eshan, Yunnan, China | NAS 0648576 | EU407549 | EU301742 | EU301740 | GQ265147 | GQ265196 | GQ265097 | GQ265243 |
| *D. gracillima* Miq. | Mt. Lushan, Jiangxi, China | NAS 0648577 | DQ974190 | DQ841315 | DQ408164 | GQ265150 | GQ265201 | GQ265101 | GQ265247 |
| *D. collettii* Hook. f. var. *collettii* | Jinghong, Yunnan, China | NAS 0648578 | DQ974178 | DQ841300 | DQ408173 | GQ265141 | GQ265192 | GQ265093 | GQ265239 |
| *D. collettii* var. *hypoglauca* (Palibin) C. Pei & C. T. Ting | Mt. Hengshan, Hunan, China | NAS 0648579 | DQ974176 | DQ841319 | EF614220 | GQ265155 | GQ265205 | GQ265106 | GQ265252 |
| *D. futschauensis* Uline ex R. Knuth | Yongtai, Fujian, China | NAS 0648580 | DQ974175 | DQ841316 | DQ408166 | GQ265151 | GQ265202 | GQ265102 | GQ265248 |
| *D. spongiosa* J. Q. Xi, M. Mizuno & W. L. Zhao | Mt. Hengshan, Hunan, China | NAS 0648581 | DQ974191 | DQ841317 | DQ974194 | GQ265153 | GQ265204 | GQ265104 | GQ265250 |
| *D. banzhuana* C. Pei & C. T. Ting | Mengzi, Yunnan, China | NAS 0648582 | DQ974182 | DQ841301 | DQ408174 | GQ265167 | GQ265218 | GQ265118 | GQ265264 |
| *D. simulans* Prain & Burkill | Guilin, Guangxi, China | NAS 0648583 | EF614206 | DQ841320 | EF614217 | GQ265138 | GQ265189 | GQ265090 | GQ265236 |
| *D. esculenta* (Lour.) Burkill | Lingshui, Hainan, China | NAS 0648585 | AY956497 * | DQ841298 | AY904794 * | GQ265180 | GQ265231 | GQ265131 | GQ265277 |
| *D. esculenta* var. *spinosa* (Roxburgh ex Prain & Burkill) R. Knuth | Lingshui, Hainan, China | B. C. Wu200804023 | GQ265087 | GQ265290 | GQ265186 | GQ265181 | GQ265232 | GQ265134 | GQ265280 |
| *D. tentaculigera* Prain & Burkill | Lincang, Yunnan, China | NAS 0648465 | GQ265089 | GQ265292 | GQ265188 | GQ265183 | GQ265234 | GQ265137 | GQ265282 |
| *D. yunnanensis* Prain & Burkill | Lijiang, Yunnan, China | NAS 0648459 | EF614209 | GQ265288 | EF614221 | GQ265161 | GQ265213 | GQ265113 | GQ265259 |
| *D. subcalva* Prain & Burkill | Tianlin, Guangxi, China | NAS 0648544 | EF614208 | EF614222 | EF614214 | GQ265160 | GQ265211 | GQ265111 | GQ265257 |
| *D. subcalva* var. *submollis* (R. Knuth) C. T. Ting & P. P. Ling | Mt. Jinfo, Chongqing, China | NAS 0648545 | EF614204 | GQ265287 | EF614216 | GQ265152 | GQ265203 | GQ265103 | GQ265249 |
| *D. nitens* Prain & Burkill | Lijiang, Yunnan, China | NAS 0648586 | EF614205 | EF614223 | EF614215 | GQ265162 | GQ265214 | GQ265114 | GQ265260 |
| *D. bulbifera* L. | Jinghong, Yunnan, China | NAS 0648587 | AY956488 * | EF619352 | AY904791 * | GQ265178 | GQ265229 | GQ265132 | GQ265278 |
| *D. melanophyma* Prain & Burkill | Mengzi, Yunnan, China | NAS 0648548 | EF614210 | DQ841303 | DQ408176 | GQ265143 | GQ265194 | GQ265095 | GQ265241 |
| *D. kamoonensis* Kunth | Mengzi, Yunnan, China | NAS 0648549 | EF028332 | DQ841302 | DQ408175 | GQ265142 | GQ265193 | GQ265094 | GQ265240 |
| *D. delavayi* Franchet | Kunming, Yunnan, China | NAS 0648550 | GQ265085 | GQ265284 | DQ974196 | GQ265148 | GQ265199 | GQ265100 | GQ265246 |
| *D. menglaensis* H. Li | Jinghong, Yunnan, China | NAS 0648461 | GQ265086 | GQ265285 | GQ265185 | GQ265146 | GQ265198 | GQ265099 | GQ265245 |
| *D. pentaphylla* Linnaeus | Guilin, Guangxi, China | NAS 0648551 | AY972483 * | DQ841327 | AF307470 * | GQ265140 | GQ265191 | GQ265092 | GQ265237 |
| *D. esquirolii* Prain & Burkill | Longzhou, Guangxi, China | NAS 0648552 | DQ974177 | DQ841322 | DQ408168 | GQ265139 | GQ265190 | GQ265091 | GQ265238 |
| *D. hispida* Dennstedt. | Longzhou, Guangxi, China | NAS 0648553 | AY957589 * | DQ841323 | AF307463 * | GQ265145 | GQ265197 | GQ265098 | GQ265244 |

**Table 1.** *Cont.*

| Species | Locality | Voucher | GenBank Accession No. | | | | | | |
|---------|----------|---------|-------|-------|-------|-----------|-----------|-------|-------|
| | | | *mat*K | *trn*L-F | *rbc*L | *psb*A-*trn*H | *rpl*36-*rps*8 | *nad*1 | *rps*3 |
| *D. aspersa* Prain & Burkill | Mengzi, Yunnan, China | NAS 0648588 | EF614211 | DQ841304 | EF614213 | GQ265168 | GQ265219 | GQ265119 | GQ265265 |
| *D. polystachya* Turczaninow | Jurong, Jiangsu, China | NAS 0648589 | EF028331 | DQ841313 | DQ408181 | GQ265144 | GQ265195 | GQ265096 | GQ265242 |
| *D. japonica* Thunberg | Lin'an, Zhejiang, China | NAS 0648590 | DQ974183 | DQ841307 | AF307457 * | GQ265170 | GQ265221 | GQ265121 | GQ265267 |
| *D. cirrhosa* Loureiro | Longzhou, Guangxi, China | NAS 0648591 | EF028329 | DQ841324 | AY904792 * | GQ265158 | GQ265209 | GQ265109 | GQ265255 |
| *D. cirrhosa* var. *cylindrica* C. T. Ting & M. C. Chang | Mt. Diaoluo, Hainan, China | NAS 0648592 | DQ974189 | DQ841314 | DQ408184 | GQ265179 | GQ265230 | GQ265133 | GQ265279 |
| *D. wallichii* J. D. Hooker | - | - | AY973830 * | - | AY939888 * | - | - | - | - |
| *D. glabra* Roxburgh | Longzhou, Guangxi, China | NAS 0648593 | AY956501 * | DQ841321 | AF307456* | GQ265157 | GQ265208 | GQ265108 | GQ265254 |
| *D. fordii* Prain & Burkill | Guilin, Guangxi, China | NAS 0648594 | EF028333 | DQ841299 | DQ974195 | GQ265156 | GQ265207 | GQ265107 | GQ265253 |
| *D. persimilis* Prain & Burkill | Mingxi, Fujian, China | NAS 0648595 | DQ974193 | DQ841328 | DQ408165 | GQ265175 | GQ265226 | GQ265127 | GQ265273 |
| *D. exalata* C. T. Ting & M. C. Chang | Tianlin, Guangxi, China | NAS 0648596 | EF028330 | DQ841325 | DQ408170 | GQ265159 | GQ265210 | GQ265110 | GQ265256 |
| *D. alata* Linnaeus | Jinghong, Yunnan, China | NAS 0648597 | AB040208 * | DQ841331 | AY667098 * | GQ265165 | GQ265216 | GQ265116 | GQ265262 |
| *D. decipiens* J. D. Hooker | Jinghong, Yunnan, China | NAS 0648598 | DQ974181 | DQ841329 | AF307454 * | GQ265166 | GQ265217 | GQ265117 | GQ265263 |
| *D. composita* Hemsl. | Jinghong, Yunnan, China | NAS 0648405 | DQ974180 | DQ841330 | DQ408172 | GQ265164 | GQ265215 | GQ265115 | GQ265261 |
| *D. sansibarensis* Pax | Botanical Garden Regen Germany | Y. F. Zhou200403004 | DQ974187 | DQ841296 | AY939883 * | GQ265177 | GQ265228 | GQ265129 | GQ265275 |
| *D. caucasica* Lipsky | Lyon, France | NAS 0648584 | DQ974188 | DQ841297 | DQ408182 | - | - | GQ265130 | GQ265276 |
| *D. elephantipes* Engl. | South Africa | N. Sheng200511014 | AY956496 * | DQ841306 | AF307461 * | GQ265176 | GQ265227 | GQ265128 | GQ265274 |
| *D. villosa* L. | America | NAS 0648463 | - | GQ265286 | DQ006092 * | GQ265149 | GQ265200 | GQ265126 | GQ265272 |
| *Tacca chantieri* André | - | - | AY973837 * | FJ194472 * | AJ235810 * | EF590744 * | - | DQ786152 * | - |

* Sequences obtained from Genbank.

### 2.2. DNA Extraction, Amplification and Sequencing

The total genomic DNA was extracted from fresh or silica-gel dried leaves following a modified CTAB method [23] and stored at −20 °C before amplification. Five chloroplast markers, *mat*K, *rbc*L, *trn*L-F, *psb*A-*trn*H and *rpl36-rps8*, plus two mitochondrial markers, *nad1* and *rps3*, were chosen, taking evolutionary rate and ease of sequencing into account. The primers used for *mat*K and *rbc*L amplification followed Gao et al.'s recommendations [24]. The amplification of *trn*L-F used primers described by Taberlet et al. [25]. The primers for *psb*A-*trn*H and *rpl36-rps8* were newly designed based on homologous sequences of *D. elephantipes* (L'Hér.) Engl. on GenBank. The primers for the *nad1* gene were designed based on the sequence of *Oryza sativa* Linn. The *rps3* gene was amplified according to Laroche and Bousquet [26]. The sequences of all primers used in this study are detailed in Table 2.

**Table 2.** Markers and primers used in this study.

| Marker | Primer Name | Direction | Primer Sequence (5′ to 3′) |
|---|---|---|---|
| *mat*K | *mat*K MF | forward | ATT TGC GAT CTA TTC ATT CAA T |
| | *mat*K MR | reverse | TGA GAT TCC GCA GGT CAT T |
| *rbc*L | *rbc*L m3 | forward | TAT CTT AGC GCC ATT CCG AGT A |
| | *rbc*L m4 | reverse | CGC GGA TAA TTT CAT TAC CTT C |
| *trn*L-F | *trn*L-F c | forward | CGA AAT CGG TAG ACG CTA CG |
| | *trn*L-F f | reverse | ATT TGA ACT GGT GAC ACG AG |
| *psb*A- *trn*H | *psb*A F1 | forward | AAT GCT CAC AAC TTY CCT CTA |
| | *trn*H R1 | reverse | CCA CTG CCT TGA TCC ACT TG |
| *rpl36- rps8* | *rpl36* F1 | forward | TTA CCC YTG TCT YTG TTT ATG |
| | *rps8* R1 | reverse | CTA CGA GAR GGT TTT ATT GAA |
| *nad1* | *nad1* F1 | forward | CCT TGT GAG CAC GTT TGG AT |
| | *nad1* R1 | reverse | GAC AAT CTC ACT CGA ATT ACA G |
| *rps3* | *rps3* F1 | forward | GTT CGA TAC GTC CAC CTA C |
| | *rps3* R1 | reverse | GTA CGT TTC GGA TAT RGC AC |

The PCR reaction mixture contained 40 ng of genomic DNA template, 2.5 mmol/L MgCl$_2$, 1 × Mg-free DNA polymerase buffer, 0.12 mmol/L dNTPs, 0.3 mmol/L of each primer, 1 U Taq DNA polymerase and water added accordingly to a final volume of 50 µL. The PCR program for *mat*K, *rbc*L, *trn*L-F, *nad1* and *rps3* was as follows: a 3 min premelt at 94 °C, followed by 35 cycles of 45 s denaturation at 94 °C, 30 s annealing at 58 °C and a 1.5 min extension at 72 °C, plus a final extension of 5 min at 72 °C. For *psb*A-*trn*H and *rpl36-rps8*, the PCR reaction included an initial denaturation at 94 °C for 3 min, followed by 35 cycles of denaturation at 94 °C for 30 s, annealing at 54 °C for 30 s, extension at 72 °C for 80 s and a final extension at 72 °C for 5 min. The PCR products were examined electrophoretically using 0.8–1.2% agarose gels and purified using a TIAN gel Midi Purification Kit (TIANGEN Biotech, Beijing, China). The purified products were sequenced using the same primer pairs with PCR. All of the sequences obtained were subjected to a BLAST (https://blast.ncbi.nlm.nih.gov/Blast.cgi (accessed on 15 April 2020)) search to detect contamination and nonspecific amplification. After confirmation, the newly generated sequences were submitted to GenBank.

### 2.3. Phylogenetic Analyses

The assembled sequences were aligned using MAFFT v.7 [27] with default settings and then adjusted manually for accuracy in Geneious R9 9.1.8 (https://www.geneious.com (accessed on 4 July 2020)). Character gaps were treated as missing data. Phylogenetic relationships were reconstructed using maximum likelihood (ML), maximum parsimony (MP) and Bayesian inference (BI) approaches. ML analysis was firstly performed to build a single-gene tree in online CIPRES Science Gateway v.3.3 (http://www.phylo.org/ (accessed on 18 Octobor 2020)) [28], using RAxML-HPC v8.2.10 [29,30] under the GTR + G DNA

substitution model [31]. One hundred rapid bootstrap replicates were generated for each single gene matrix [32]. Since there was no strong conflict among individual markers, we conducted further ML analysis using a concatenated matrix of all seven loci under the GTR + G model as recommended by jModelTest v2.1 [33]. Bootstrap analyses were used to evaluate the support for each clade with 1000 bootstrap replicates. MP analysis was conducted in PAUP* version 4.0b10 [34]. All characters were equally weighted. Trees were inferred using the heuristic search option with tree bisection probabilities (TBR) swapping and 1000 replicates of random addition. Ten trees were held in each step during stepwise addition. The maximum number of trees was set to 10,000, and all parsimonious trees were saved. The tree length (TL), consistency index (CI), retention index (RI) and rescaled consistency index (RC) were calculated for each maximum parsimony tree. The bootstrap results were summarized in a 50% majority-rule consensus cladogram. BI analysis was implemented using MrBayes version 3.2.6. [35]. A Markov Chain Monte Carlo (MCMC) analysis was run with two independent chains with a random starting tree for 100 million generations, sampling one tree every 1000 generations. The first 25% generations were discarded as burn-in, and the remaining trees were used to construct a consensus tree with a 50% majority rule and obtain the posterior probabilities.

### 2.4. Character Coding and Ancestral State Reconstruction

Four characters closely related to vegetative or sexual reproduction were selected for ancestral character state reconstruction, namely bulbil formation, inflorescence structure, floral color and opening degree of the perianth. Morphological data were obtained from direct observations and the literature. The evolutionary history of each of the four characters was traced over the Bayesian 50% majority-rule tree using MP approaches available in Mesquite 3.5.1 [36]. The character states were treated as unordered and equally weighted. Morphological characters and their states were coded as follows: a. bulbil absence (0), presence (1); b. inflorescence umbel (0), spike (1), raceme (2) or panicle (3); c. floral color dark (e.g., violet, purplish red, orange) (0) or light (e.g., white, pale yellow, green) (1); d. perianth opening degree open (0), half-closed (1) or closed (2). For species in which the floral color changed during blooming, we used the color at the mature stage as the character state. The resulting codes for the sampled species are summarized in Table 3.

**Table 3.** Data matrix of morphological characters used in this study.

| Species | | Character | | | |
|---|---|:---:|:---:|:---:|:---:|
| | | a | b | c | d |
| *Dioscorea* sect. *Stenophora* | *D. nipponica* | 0 | 1 | 1 | 0 |
| | *D. nipponica* subsp. *rosthornii* | 0 | 2 | 1 | 0 |
| | *D. althaeoides* | 0 | 3 | 1 | 0 |
| | *D. tokoro* | 0 | 3 | 1 | 0 |
| | *D. zingiberensis* | 0 | 1 | 0 | 0 |
| | *D. sinoparviflora* | 0 | 1 | 0 | 0 |
| | *D. deltoidea* | 0 | 1 | 1 | 0 |
| | *D. panthaica* | 0 | 3 | 1 | 0 |
| | *D. biformifolia* | 0 | 3 | 1 | 0 |
| | *D. gracillima* | 0 | 1 | 1 | 0 |
| | *D. collettii* | 0 | 1 | 1 | 0 |
| | *D. collettii* var. *hypoglauca* | 0 | 1 | 1 | 0 |
| | *D. futschauensis* | 0 | 3 | 0 | 0 |
| | *D. spongiosa* | 0 | 3 | 1 | 0 |
| | *D. banzhuana* | 0 | 3 | 1 | 0 |
| | *D. simulans* | 0 | 2 | 0 | 0 |
| | *D. caucasica* | 0 | 1 | 1 | 0 |
| | *D. villosa* | 0 | 1 | 1 | 1 |

**Table 3.** *Cont.*

| Species | | Character | | | |
|---|---|:---:|:---:|:---:|:---:|
| | | **a** | **b** | **c** | **d** |
| *D.* sect. *Combilium* | *D. esculenta* | 0 | 1 | 1 | 0 |
| | *D. esculenta* var. *spinosa* | 0 | 1 | 1 | 0 |
| *D.* sect. *Shannicorea* | *D. tentaculigera* | 1 | 1 | 1 | 0 |
| | *D. yunnanensis* | 0 | 1 | 1 | 0 |
| | *D. subcalva* | 0 | 1 | 1 | 0 |
| | *D. subcalva* var. *submollis* | 0 | 1 | 1 | 0 |
| | *D. nitens* | 0 | 1 | 1 | 0 |
| *D.* sect. *Opsophyton* | *D. bulbifera* | 1 | 1 | 0 | 1 |
| *D.* sect. *Botryosicyos* | *D. melanophyma* | 1 | 2 | 1 | 2 |
| | *D. kamoonensis* | 1 | 2 | 1 | 2 |
| | *D. delavayi* | 1 | 2 | 1 | 2 |
| | *D. menglaensis* | 1 | 3 | 1 | 2 |
| | *D. pentaphylla* | 1 | 3 | 1 | 1 |
| | *D. esquirolii* | 1 | 2 | 1 | 1 |
| *D.* sect. *Lasiophyton* | *D. hispida* | 0 | 3 | 1 | 2 |
| *D.* sect. *Enantiophyllum* | *D. aspersa* | 0 | 1 | 1 | 2 |
| | *D. polystachya* | 1 | 1 | 1 | 2 |
| | *D. japonica* | 1 | 1 | 1 | 2 |
| | *D. cirrhosa* | 1 | 1 | 1 | 2 |
| | *D. cirrhosa* var. *cylindrica* | 1 | 3 | 1 | 2 |
| | *D. wallichii* | 0 | 3 | 1 | 2 |
| | *D. glabra* | 1 | 3 | 1 | 2 |
| | *D. fordii* | 1 | 3 | 1 | 2 |
| | *D. persimilis* | 1 | 3 | 1 | 2 |
| | *D. exalata* | 1 | 3 | 1 | 2 |
| | *D. alata* | 1 | 3 | 1 | 2 |
| | *D. decipiens* | 1 | 3 | 1 | 2 |
| *D.* sect. *Apodostemon* | *D. composita* | 0 | 1 | 1 | 1 |
| *D.* sect. *Macroura* | *D. sansibarensis* | 1 | 1 | 1 | 2 |
| *D.* sect. *Testudinaria* | *D. elephantipes* | 0 | 2 | 0 | 0 |
| outgroup | *Tacca chantieri* | 0 | 0 | 0 | 0 |

## 3. Results

### 3.1. Phylogenetic Analyses

According to our analyses, the chloroplast segment *trn*L-F exhibited the highest variability among the seven molecular markers, followed by *mat*K and *nad1* (see Table 4). The final concatenated matrix of all seven markers consisted of 7218 base pairs containing 1157 variable sites and 614 parsimony-informative sites. The molecular reconstruction based on the combined matrix gave rise to a highly resolved phylogenetic tree with moderate to strong support. There was no significant incongruence in topology or support values among trees generated from different methods. Thus, only the ML tree is presented in Figure 2.

**Table 4.** Informative parameters for the seven molecular markers.

| DNA Region | Aligned Length (bp) | Variable Site (bp/%) | Informative Site (bp/%) | Tree Length | *CI* | *RI* |
|:---:|:---:|:---:|:---:|:---:|:---:|:---:|
| *mat*K | 1032 | 237/23.0 | 163/15.8 | 317 | 0.852 | 0.959 |
| *rbc*L | 1142 | 134/11.7 | 84/7.4 | 220 | 0.664 | 0.867 |
| *trn*L-F | 939 | 280/29.8 | 80/8.5 | 385 | 0.816 | 0.905 |
| *psb*A-*trn*H | 344 | 113/32.8 | 82/23.8 | 160 | 0.850 | 0.955 |
| *rpl*36-*rps*8 | 857 | 111/13.0 | 64/7.5 | 142 | 0.817 | 0.949 |
| *nad*1 | 1478 | 207/14.0 | 125/8.5 | 246 | 0.902 | 0.949 |
| *rps*3 | 1426 | 75/5.3 | 19/1.3 | 89 | 0.876 | 0.929 |
| 7 DNA | 7218 | 1157/16.0 | 617/8.5 | 1640 | 0.782 | 0.913 |

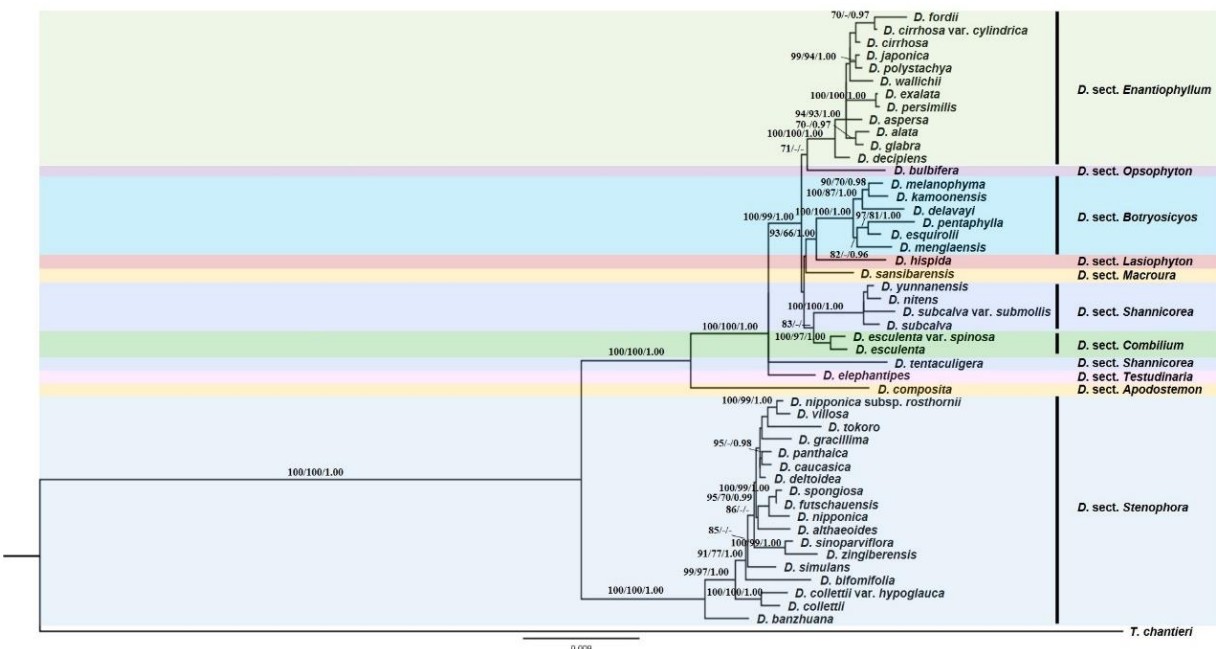

**Figure 2.** The maximum likelihood tree using the combined data from chloroplast *mat*K, *rbc*L, *trn*L-F, *psb*A-*trn*H, *rpl36-rps8*, as well as mitochondria *nad1* and *rps3*. Branches are labeled with maximum likelihood bootstraps higher than 70%, parsimony bootstrap proportions higher than 50% and Bayesian posterior probabilities more than 0.95.

All sampled *Dioscorea* species clustered in a monophyletic group consisting of two major clades with strong support. In accordance with previous studies, *D.* sect. *Stenophora* was the first diverging clade, sister to the other clade that included all the remaining species (BS = 100). The species of *D.* sect. *Botryosicyos* was grouped into a monophyletic clade (BS = 100). This clade was further divided into two subclades, one consisting of *D. pentaphylla* L., *D. esquirolii* Prain et Burkill and *D. menglaensis* H. Li and the other including *D. melanophyma* Prain et Burkill, *D. kamoonensis* Kunth and *D. delavayi* Franch. The only species of *D.* sect. *Lasiophyton*, *D. hispida* Dennst, was sister to *D.* sect. *Botryosicyos* (BS = 93). The position of *D. tentaculigera* Prain et Burkill remained unresolved, while other species of *D.* sect. *Shannicorea* clustered together in a well-supported clade (BS = 100). The two samples of *D.* sect. *Combilium* grouped together, and they were revealed as sisters to the Shannicorea clade with moderate support (BS = 83). All accessions of *D.* sect. *Enantiophyllum* formed a well-supported monophyletic clade (BS = 100). The species of *D.* sect. *Opsophyton* (*D. bulbifera*) was positioned as a sister to *D.* sect. *Enantiophyllum* with moderate support (BS = 71).

Some taxonomic treatments at the species level were supported by the phylogenetic tree. *Dioscorea collettii* Hook. f. and *D. collettii* var. *hypoglauca* (Palibin) C. T. Ting grouped together, while *D. esculenta* (Lour.) Burkill and *D. esculenta* var. *spinosa* (Roxburgh ex Prain & Burkill) R. Knuth were resolved as the closest relatives to each other. However, some varieties or subspecies of one species did not form a monophyletic group. For instance, *D. nipponica* Makino and *D. nipponica* subsp. *rosthornii* (Prain & Burkill) C. T. Ting, and *D. subcalva* Prain et Burkill and *D. subcalva* var. *submollis* (R. Knuth) C. T. Ting & P. P. Ling, did not group together.

*3.2. Ancestral Character State Analyses*

The ancestral state of four reproductive features of the *Dioscorea* species was reconstructed based on MP methods. As for character "a" the absence or presence of bulbils at the axil), state "0" (absent) was suggested to be the plesiomorphic state for *Dioscorea* (Figure 3). No species of the outgroup *Tacca* and *D.* sect. *Stenophora*, the earliest diverging group in *Dioscorea*, produces bulbils. The trait distribution in the phylogenetic tree suggests

that bulbils appeared after the divergence of *D.* sect. *Stenophora,* but were subsequently lost several times independently, in *D.* sect. *Combilium*, *D.* sect. *Lasiophyton* and several species in *D.* sect. *Enantiophyllum* (*D. wallichii* Hook. f., *D. aspersa* Prain et Burkill). Bulbils persisted in most species in *D.* sect. *Botryosicyos*, *D.* sect. *Opsophyton* and *D.* sect. *Enantiophyllum*.

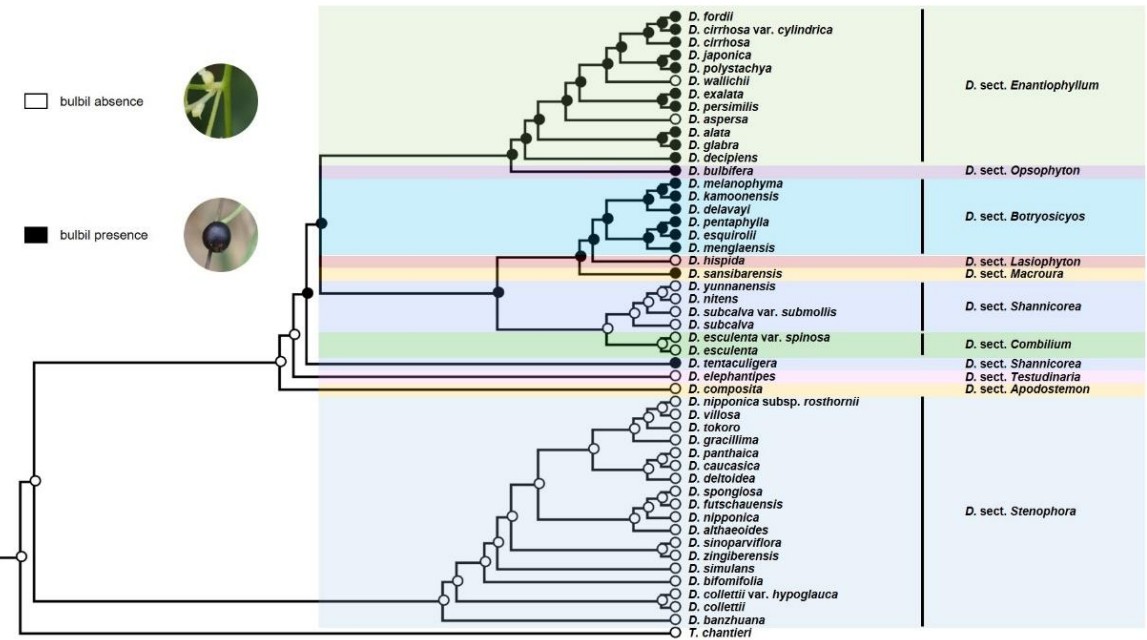

**Figure 3.** Ancestral character state reconstruction for bulbils using the parsimony method.

The ancestral state of the inflorescence architecture was inferred to be a spike (Figure 4). Racemes and panicles were inferred to be derived states, with the former occurring sporadically in different clades and the latter occurring mainly in *D.* sect. *Botryosicyos* and *D.* sect. *Enantiophyllum*. Spike occurred as a homoplasy in *D.* sect. *Shannicorea,* while other clades did not show a consistent type of male inflorescence.

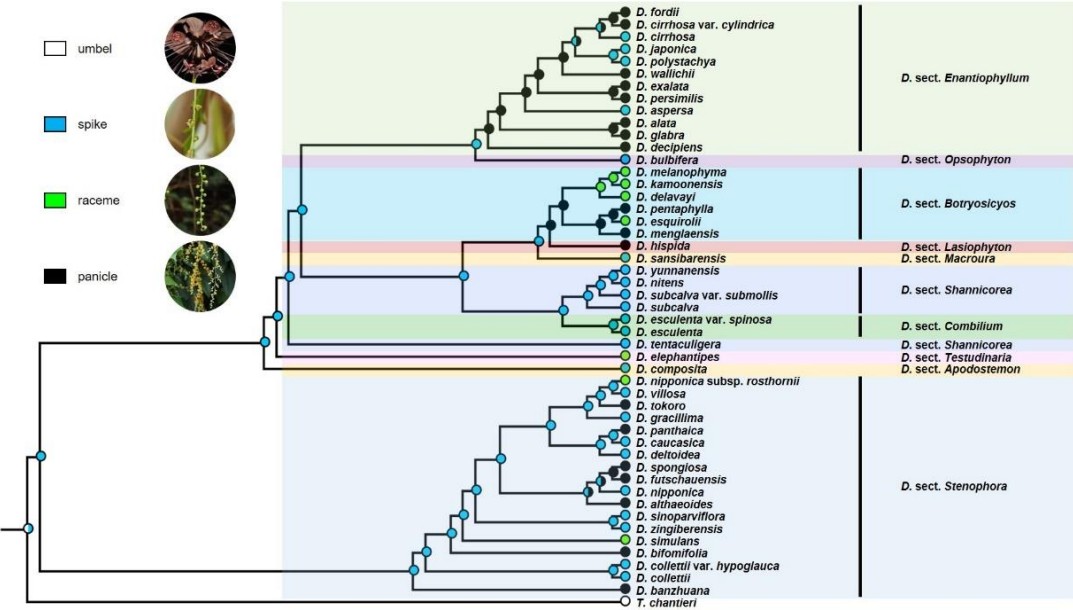

**Figure 4.** Ancestral character state reconstruction for the inflorescence architecture using the parsimony method.

The light color of the male flowers was uncovered as the ancestral state of *Dioscorea* (Figure 5). The dark color evolved independently several times in *D*. sect. *Stenophora* and *D*. sect. *Opsophyton*. Most species with dark colored male flowers occurred in *D*. sect. *Stenophora*. There were also some species in *D*. sect. *Shannicorea* and *D*. sect. *Enantiophyllum* that possessed male flowers that were purple in color or had a brown stripe on them. This distribution probably reflects multiple convergence events in response to similar pollination strategies.

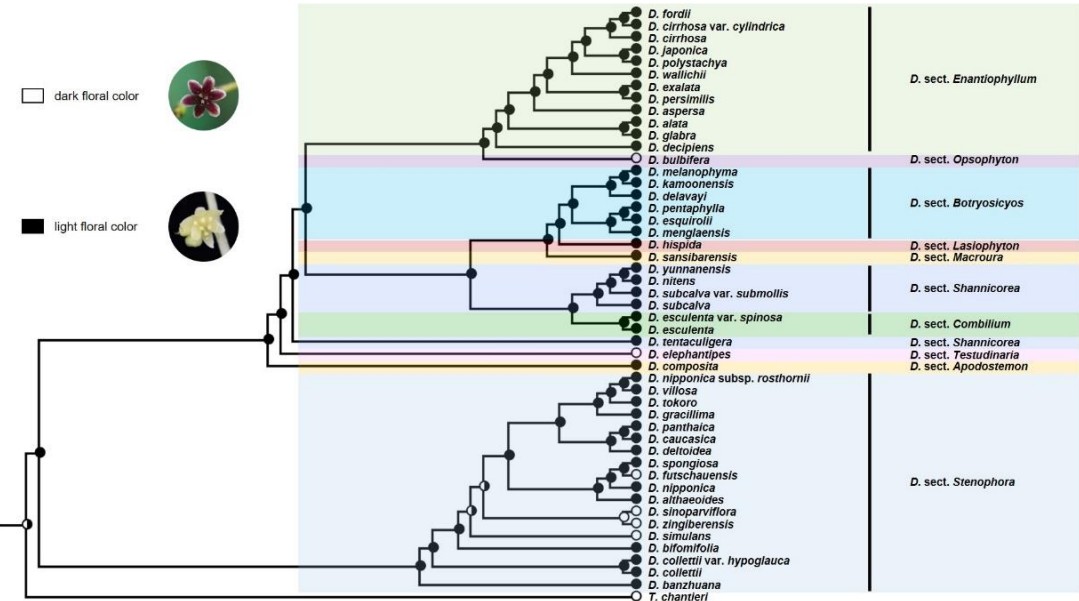

**Figure 5.** Ancestral character state reconstruction for the floral color using the parsimony method.

The male flowers of *Dioscorea* ancestors were suggested to open fully, and this state was retained in *D*. sect. *Stenophora* and *D*. sect. *Shannicorea* (Figure 6). The perianth of male flowers was closed to some degree in the ancestor of the clade encompassing *D*. sect. *Enantiophyllum*, *D*. sect. *Opsophyton*, *D*. sect. *Botryosicyos*, *D*. sect. *Lasiophyton* and *D*. sect. *Combilium*. Male flowers of *D*. sect. *Botryosicyos* and *D*. sect. *Enantiophyllum* were generally closed.

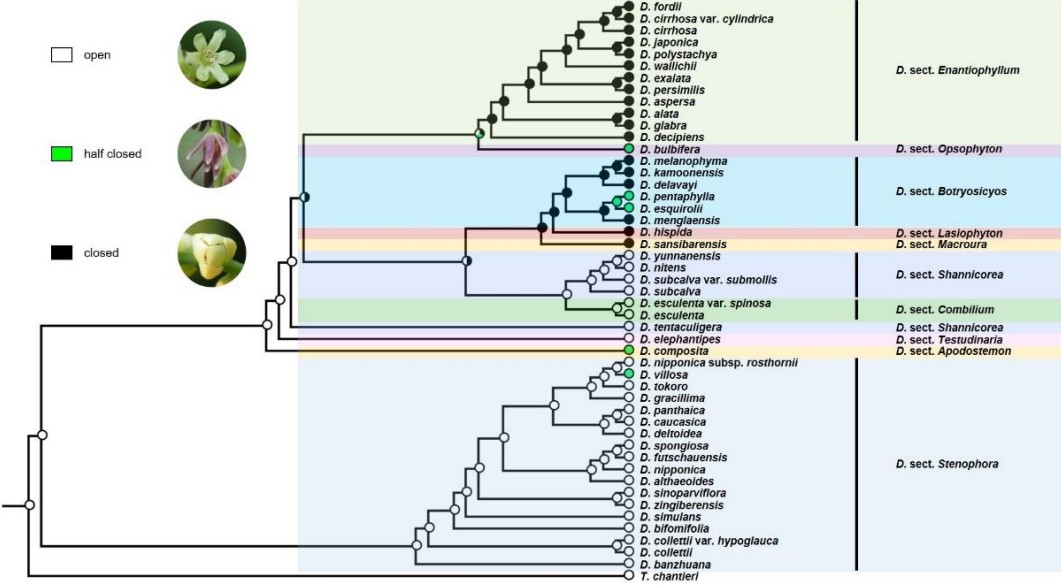

**Figure 6.** Ancestral character state reconstruction for the perianth opening degree using the parsimony method.

## 4. Discussion

### 4.1. Evolution of Organs for Vegetative Propagation and Diversity in the Reproductive Strategy

Vegetative propagation is a crucial mechanism that ensures plants maintain and spread their population, allowing plants to survive unfavorable conditions for sexual reproduction [37]. *Dioscorea* species possess multiple organs involved in vegetative propagation, including rhizomes, horizontal or vertical tubers and aerial bulbils, as a result of adaptation to various environmental conditions. In the early diverged *D*. sect. *Stenophora*, the underground parts consisted of perennial branched horizontal rhizomes. Perennial to annual starchy tubers took the place of rhizomes in subsequently evolved lineages to undertake the function of asexual reproduction. Rhizomatous species are supposed to be more adaptive to long growing seasons in shaded habitats [38], whereas tubers can evade destruction by animals and allow adaptation to warm and humid climates [39]. The possession of tubers was suggested to play an important role in species diversification of *Dioscorea* [2]. However, tubers are mainly storage organs for plants to live through winter and germinate in for the next growing season; the reproductive efficiency and dispersal ability of tubers are limited.

The emergence of aerial bulbils in some *Dioscorea* species greatly changed the propagation patterns of these species. The bulbil of *Dioscorea* is a minor storage organ compared to the underground tuber, arising from the axils of leaves or inflorescences and varying in form and size in different species [40]. On the basis of morphological and anatomical studies, bulbils are interpreted as modified axillary branches and miniature reproductions of the rhizome [11,41]. Functioning as a means of vegetative propagation, bulbils are released from the plants after maturation and then dispersed by gravity or by water in streams, sometimes over long distances, like seeds. Bulbils can easily germinate and grow independently, giving rise to large numbers of progeny. Because of these advantageous characteristics, species that produce bulbils are usually more vigorous and more competitive than species that reproduce only by seeds, as observed in *Allium* (Alliaceae) [42] and *Butomus* (Butomaceae) [43].

As mentioned above, the vegetative propagation strategies vary among the different lineages of *Dioscorea*, and the production of bulbils is a derived character in most clades of this genus. This may have greatly influenced the expansion of the *Dioscorea* species. In *Poa alpina* L. (Poaceae), bulbil-producing plants are better adapted to higher elevations compared with seed-producing plants. Therefore, this species is able to occupy a range of ecological niches by means of different reproductive modes [44]. Bulbils in *Fritillaria* (Liliaceae) are suspected to be an adaptation to underground dispersal [38]. In *Dioscorea*, the rhizomatous taxa are mainly distributed in South and East Asia, and many of them are endemic species only found in restricted areas, while the species that produce bulbils occupy a wide range of habitats worldwide [2]. *D. bulbifera*, the only wild species exhibiting worldwide distribution in *Dioscorea*, produces huge numbers of bulbils compared to other species. Bulbils of *D. sansibarensis* Pax are buoyant and spread through water flow, achieving wide distribution in African valleys [11]. Here, it is postulated that the production of bulbils improved the adaptive ability of *Dioscorea* species and promoted their expansion.

### 4.2. Evolution of Floral Characters and Their Relation to Sexual Propagation

Flower evolution is strictly linked to pollination strategy [45,46]. *Dioscorea* is predominantly dioecious and allogamous; thus sexual propagation in this genus relies on the outcrossing process. Moreover, the sticky nature of *Dioscorea* pollen grains prevents transport by wind, and, consequently, the flowers of *Dioscorea* species are pollinated mainly by insects [47]. The diverse flower exhibition characteristics of this genus were most likely shaped by pollinator-mediated selection.

According to our analyses, spikes are reconstructed as the plesiomorphy of *Dioscorea*, while panicles appear mainly in *D*. sect. *Enantiophyllum* as a derived inflorescence type. This contrasts with the opinion presented by Burkill [11], who suggested that spikes and racemes evolved from panicles. The distribution patterns of inflorescence types among species do not mirror phylogenetic relationships, with closely related lineages displaying

different types, except *D*. sect. *Shannicorea*. Such a pronounced polymorphism might have been driven by pollinators. Inflorescence architecture is related to the mode and efficiency of pollination. For instance, changes in inflorescence architecture were associated with the transition from biotic (insect) to abiotic (wind) pollination in *Schiedea salicaria* Hbd. (Caryophyllaceae) [48], and affected pollinator behavior and mating success in *Spiranthes sinensis* (Pers.) Ames (Orchidaceae) [49]. Panicles are made of multiple spikes or racemes to enlarge the volume of the flower and enhance the attraction of pollinators. They possess two to three times the number of flowers and amount of pollen as spikes or racemes of the same length. The function of panicles in insect attraction compared with other types of inflorescences is worthy of investigation in *Dioscorea*.

In our results, most species of *Dioscorea* possess male flowers of light color, and dark color originated several times independently in different lineages. It is well known that floral color is among the most important visual signals in pollinator attraction. For example, hawkmoth and hummingbird pollinators prefer yellow and red morphs of *Mimulus aurantiacus* Curtis, respectively [50]. The reflectance spectrum of the dark perianth generated by UV-light is recognizable to insects [51]. Dark color and light color perianth could be visually discriminated by insects and thus will affect the visiting choice of different pollinators [52]. An alternative explanation for floral color polymorphisms among closely related species is that color divergence evolves in response to interspecific competition for pollinators as a means to decrease interspecific pollinator movements [53]. Indeed, the color of perianth in *Dioscorea* includes white, yellow, green, orange, red and purple, sometimes changing from one to another during flower development. How these variations of floral colors influence the category and behavior of the pollinators will be an interesting subject to investigate.

The evolutionary trend of the perianth opening degree in *Dioscorea* is from wholly opened to nearly closed. This character may also influence the category of pollinators. As discovered in the study of *Merianieae* (Melastomataceae), corollas of most buzz-bee syndrome species are widely open, forming bowl-shaped flowers, whereas they are more closed and form urceolate to pseudo-campanulate flowers in vertebrate-pollinated species [54]. The same was also observed in *Erica* (Ericaceae) [55] and *Ruellia* (Acanthaceae) [56]. Zhao et al. [57] reported that *D. nipponica* subsp. *rosthornii* is pollinated by halictids. In *D*. sect. *Enantiophyllum* species, male panicles were found usually to be made up of closed flowers that restrict visitations only to small insects. The pollinators of species such as *D. rotundata* Poir., *D. japonica* Thunb. and *D. polystachya* Turcz. were purported to be thrips [47,58–60]. Therefore, the evolution of male flower opening degree in *Dioscorea* may be a result of pollinator specialization.

Actually, as reported in previous studies, the pollinators of *Dioscorea* species include *Coleoptera*, *Diptera*, *Hymenoptera*, *Hemiptera*, *Thripidae*, *Thysanoptera*, *Halictus* and *Andrena*. [47,60]. The diversification of floral morphological characters in this genus is thought to be promoted by pollination-associated adaptations. Intensive investigation of the pollination strategies in other lineages is needed to better understand the adaptive evolution of reproduction in *Dioscorea*.

### 4.3. Infrageneric Relationships and Taxonomic States of Several Species

The phylogenetic framework reconstructed in this study was consistent with those obtained in previous studies [2,12,14,15]. The relationships indicated by molecular information were generally consistent with infrageneric division by traditional systematics.

The monophyly of *Dioscorea* sect. *Stenophora* has been certified in a number of molecular studies using various datasets. This is also supported by morphological evidence since species of *D*. sect. *Stenophora* show underground organ type, chromosome number and pollen type that set them apart from other lineages of *Dioscorea*. As indicated in this study, no species of *D*. sect. *Stenophora* generated bulbils, and they all had male flowers with fully open perianths, thus giving further support to the isolated position of this section within the *Dioscorea* genus.

Most species of *Dioscorea* sect. *Lasiophyton* have been transferred into *D.* sect. *Botryosicyos* [22]. However, there are clear morphological differences between these two taxa. Species of *D.* sect. *Lasiophyton* have compound leaves with three palmate leaflets that are also palmately veined, and all six stamens of the male flower are fertile; in contrast, in *D.* sect. *Botryosicyos*, leaves usually have more than three leaflets, are pinnately veined, and have three stamens alternating with three staminodes. In this study, the only Chinese species of *D.* sect. *Lasiophyton*, *D. hispida*, was resolved as the sister lineage to the *Botryosicyos* clade, supporting the exclusion of this species from *D.* sect. *Botryosicyos* by Ding and Gilbert [22].

*Dioscorea tentaculigera* was assigned to *D.* sect. *Shannicorea* by Burkill [11]. Its male flowers are light in color and fully open. However, *D. tentaculigera* produces bulbils, and its male flowers are sessile, which is an obvious departure from other species in *D.* sect. *Shannicorea*. The phylogenetic position of *D. tentaculigera* was not resolved in our analyses, as has been the case in previous studies (e.g., [2,12–14]). Maurin et al. [20] placed this species outside of the *Enantiophyllum* clade based on six cpDNAs; Soto Gomez et al. [61] positioned it as a sister to the Mediterranean clade based on 260 nuclear genes; and Noda et al. [15] suggested that it should be treated as a distinct section according to four cpDNAs. More evidence from molecular and morphological data is needed to draw a conclusion about the correct classification of this species.

Accessions of *Dioscorea* sect. *Enantiophyllum* were found to cluster into a highly supported monophyletic clade. There are several synapomorphies of this group: (i) stem twining to the right, (ii) leaves generally opposite, (iii) seeds winged all round, (iv) bulbils normally present, (v) male inflorescence panicles and (vi) perianth white to yellow and nearly closed when blooming. The limits of this section were found to be reasonable for a natural lineage [2,12,14,15,17].

The phylogenetic analyses also shed light on the taxonomy of certain species. *Dioscorea nipponica* subsp. *rosthornii* was separated from *D. nipponica* subsp. *nipponica* by a cork layer of rhizomes and male flowers that were pedicellate [62], as well as a difference in chromosome number [24,39]. However, they have not been resolved as sister groups in any molecular phylogenetic trees generated to date [12,14,24]. A similar situation was observed with *D. subcalva* var. *submolis*, which showed differences from *D. subcalva* var. *subcalva* that included sparse leaf blades and longer infructescences. We noticed that these characters varied in a continuous range between individuals among populations. The relationships between these two varieties and other species of *D.* sect. *Shannicorea* remain unresolved by molecular analyses [15,17]. Further investigation with sufficient samples may help us understand the internal relationships within this section and the circumscription of the species.

## 5. Conclusions

This study provides the first phylogenetic analyses focusing on the evolution of four main reproductive traits in *Dioscorea*. The development of bulbils in late-diverged lineages of *Dioscorea* has diversified the mode of vegetative propagation, which is supposed to have greatly improved reproductive efficiency and heightened plants' ability to occupy new habitats. Spikes, male flowers in light color and wholly open perianth are reconstructed as plesiomorphic in *Dioscorea*, whereas panicles, dark color flowers and nearly closed perianth are suggested derived states. The extraordinary variation of floral characters in this genus appears to be a consequence of adaptive evolution driven by pollinators. A broader sampling of *Dioscorea,* especially new-world species, together with their character information and extensive pollination studies among species with different floral, types are necessary to achieve a better understanding of the diversification mechanism of this genus.

On the other hand, this work provides a suitable phylogenetic framework for the revision of infrageneric classification and species delimitation in *Dioscorea*. Some of the morphological characters show consistency in a certain clade. Species in *D.* sect. *Stenophora* do not produce bulbils, and their male flowers are always wholly open; *D.* sect. *Shanni-*

*corea* is characterized by no bulbils, spikes, light color and wholly open flowers; *D.* sect. *Enantiophyllum* is distinguished by producing bulbils, with flowers in light color and nearly closed. According to our results, bulbils and perianth opening degree are informative taxonomic features at the section level in *Dioscorea*. The monophyletic status of several sections proposed in classical taxonomy, such as *D.* sect. *Stenophora* and *D.* sect. *Enantiophyllum,* were supported by morphological synapomorphies besides molecular data. Morphological characters analyzed in the present work can also explain the unexpected position of *D. tentaculigera* in phylogenetic trees. In addition, our results suggest the necessity of reconsideration of some infraspecific taxa as well.

**Author Contributions:** Methodology, M.C. and J.-Y.X.; software, J.-Y.X.; validation, X.S., Y.H. and Y.Z.; formal analysis, M.C. and X.S.; investigation, M.C.; resources, Y.Z. and Y.H.; data curation, M.C.; writing—original draft preparation, M.C.; writing—review and editing, X.S. and Y.H.; supervision, Y.H.; project administration, Y.H.; funding acquisition, M.C. and X.S. All authors have read and agreed to the published version of the manuscript.

**Funding:** This study was funded by the Youth Foundation of Jiangsu Province to M.C. (Grant No. BK20180316) and the Independent Research Project of Jiangsu Provincial Public Welfare Research Institute to X.Q.S. (Grant No. BM2018021-2).

**Institutional Review Board Statement:** Not applicable.

**Data Availability Statement:** Data generated in this study have been uploaded to NCBI (https://www.ncbi.nlm.nih.gov/) and GenBank accession numbers are listed in Table 1.

**Conflicts of Interest:** The authors declare no conflict of interest.

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
