# Peer review of "Evolution of Reproductive Traits and Implications for Adaptation and Diversification in the Yam Genus Dioscorea L."

_diversity, doi:10.3390/d14050349_

Round 1

Reviewer 1 Report

I have carefully read MS which was submitted for consideration in the Diversity (MDPI). The topic of the work is undoubtedly very important and topical. Dioscorea has presented a challenge to systematists for many years due to its great morphological diversity, dioecy, and small flowers. The first taxonomic treatments of significant numbers of species were those of Kunth (1850) and Uline (1898). The last complete monograph was published by Knuth (1924). Using a typically narrow species concept, he recognized ca. 600 species and divided them into four subgenera based on seed wing position, and then into 60 sections. Since 1960, the genus has been the subject of piecemeal floristic studies (e.g., Miége 1968; Milne-Redhead 1975; Tellez and Schubert 1994; N’Kounkou 1993; Miége and Sebsebe 1998; Ding and Gilbert 2000). The only complete taxonomic treatment was that of Huber (1998), in which the Knuth-Burkill system of classification was recapitulated, with all of the dioecious taxa of Dioscoreaceae included in subfamily Dioscoreoideae as ‘‘genera and genus-equivalent sections’’. However, when the classification system is studied by a contemporary systematist, it is clear that many of Knuth’s infrageneric taxa are clearly para- or even polyphyletic. It has long been considered that this taxonomy is rather temporary and requires a complete revision and that the revision should use cladistic analysis of DNA sequence data for a significant number of species of the genus.

In this study, the authors reconstruct the phylogenetic relationships between 48 Dioscorea species using five chloroplast and two mitochondrial DNA markers and proposed a well-supported framework for defining these species. The results suggested that yam bulbils evolved after the divergence of the Dioscorea sect. Stenophora from the ancestors of other species. This article is very relevant and interesting, it focuses on poorly researched issues.

This manuscript is in general well written, logically structured, well-illustrated and easy to understand. It also addresses a subject that is of great interest in the scientific community. The title clearly describes the contents of the paper. The abstract is well written. It encapsulates the entire study (a bit of introduction, aim, result and outcome). The introduction is well written as it gives a good background of the research in question. Also, the aim of the study is evident in the beginning and concluding parts. I believe that the Materials and Methods section is well structured and scientifically sound. The results are well presented, figures and tables are correct. Literature reviews in the discussion section of the manuscript are very professional.

My comments mostly relate to relatively minor issues of interpretation and writing. These comments do not influence a positive impression of the article.

Suggestions:

Line 34: Please explain why Dioscorea is the "most critical taxon in monocot systematics" - it is very important in the context of the results of this article. Is there a problem with the definition of separate species in this group of plants?

Line 94. “Taxon sampling” modify as “Sampling” or “Plant material”

Line 226 “Character evolution” -  better to say “Ancestral Character State Analyses”. Please consider combining subsections 3.1 and 3.2 into one: Phylogenetic Analyses and Species Recognition

I think that a subsection "Conclusions" should be added. Please summarize the research results and provide clear taxonomic conclusions. What have your results changed in the current classification system?

Reviewer 2 Report

An interesting paper providing useful novel information of the systematics and evolution of Dioscorea.

Lines 48-50: explain if “short branches” are phylogenetic branches or real branches

Line 233: According to Fig. 3, bulbils do occur in D. tentaguligera

Further suggestions are included in the pdf attached.
